# Systematic Surveillance and Meta-Analysis of Antimicrobial Resistance and Food Sources from China and the USA

**DOI:** 10.3390/antibiotics11111471

**Published:** 2022-10-25

**Authors:** Carlos R. Prudencio, Antonio Charlys da Costa, Elcio Leal, Chung-Ming Chang, Ramendra Pati Pandey

**Affiliations:** 1Department of Biotechnology, SRM University, Rajiv Gandhi Education City, P.S. Rai, Sonepat 131029, Haryana, India; 2Center of Immunology, Institute Adolfo Lutz, São Paulo 01246-902, Brazil; 3Instituto de Medicina Tropical, Universidade de São Paulo, São Paulo 05403-000, Brazil; 4Laboratório de Diversidade Viral, Instituto de Ciências Biológicas, Universidade Federal do Pará, Belem 66075-000, Brazil; 5Master & Ph.D. Program in Biotechnology Industry, Chang Gung University, No. 259, Wenhua 1st Rd., Guishan Dist., Taoyuan 33302, Taiwan

**Keywords:** antimicrobial resistance (AMR), antibiotics, surveillance, China, USA

## Abstract

**Highlights:**

Systematic analyzation to assess the spread of AMR bacteria prevalence in retail food products and the subsequent exposure to antibiotic resistance.Out of 13,018 food samples, 5000 samples were contaminated.Meat shows high to medium potential of AMR exposure for Gram-positive and Gram-negative foodborne pathogens.*Salmonella* and *Staphylococcus aureus* were two predominant bacteria seen in China and the USA, respectively.Multidrug resistance was detected in most of the food samples from both countries.Food samples were more resistant to β-lactams and tetracyclines.Government bodies were formed to tackle AMR from food.

**Abstract:**

Since the discovery of antibiotics in the 20th century, they have been used to fight against infections. The overuse of antibiotics in the wider environment has resulted in the emergence of multidrug-resistant bacteria. In developing countries such as China and developed countries such as the USA, there is evidence of the high pervasiveness of antibiotic-resistant infections. However, the studies on the spread of antibiotic-resistant microorganisms that inform about the consequences are limited. The aim of our study was to analyze and compare antimicrobial resistance (AMR) identified in published research papers from that found in different food sources, which were published between 2012 and December 2021, covering most retail food items. Out of 132 research papers identified, 26 papers have met our strict criteria and are included in the qualitative and quantitative analysis. The selected papers led to 13,018 food samples, out of which 5000 samples were contaminated, including 2276 and 2724 samples from China and the USA, respectively. Meat, aquatic products, milk, and eggs show high to medium potential for AMR exposure to Gram-positive bacteria such as *Staphylococcus, Enterococci*, etc. and Gram-negative foodborne pathogens such as *Campylobacter*, *Salmonella*, *Vibrio*, etc. Most of the food samples show antibiotic resistance to β-lactams, tetracycline, quinolones, and aminoglycosides. Retail food products such as meat, sea food, and some other food products, as well as AMR genetics and technically important bacteria, are proposed to be better merged with mitigation strategies and systematic One Health AMR surveillance to minimize the knowledge gaps and facilitate comprehensive AMR risk computation for the consumers.

## 1. Introduction

AMR has become a significant threat to public health worldwide. AMR is defined as the ability of bacteria, viruses, parasites, fungi, etc., to grow and spread in the presence of antimicrobial medicines to which they were previously susceptible, leading to an increase in the risk of diseases spreading to others [1]. AMR can occur due to various mechanisms, e.g., horizontal gene transfer, mutations in previously acquired genes, enzymatic degradation/hydrolyses, impermeably modified antimicrobial targets, etc. [2,3]. More antibiotic-susceptible strains have resulted in increased infection with a higher rate of morbidity, mortality, and social and economic losses. AMR development is a naturally occurring phenomenon, but it has recently increased due to the overuse of unnecessary, nonprescribed antibiotics, as well as the over-prescription of antibiotics in daily lives, which has catalyzed this phenomenon. Widespread use of antibiotics has been seen in livestock, aquaculture, and agriculture, because many of the antibiotics used in livestock are the same as the antibiotics used in humans. The World Health Organization (WHO) has long recognized the importance of a more effective and coordinated worldwide effort to combat AMR. The WHO Global Strategy for Antimicrobial Resistance Containment, published in 2001, established a framework of measures to prevent the increasing prevalence of antimicrobial-resistant microbes [4]. In the European Union (EU), it is estimated that antibiotic-resistant pathogens (ARPs) are responsible for approximately 33,000 deaths per year and 4.95 million deaths globally, with healthcare expenditures and productivity losses amounting to EUR 1.5 billion every year [5]. According to a report, by 2050, AMR will play a part in more than 10 million deaths per year, with a loss of more than 100 trillion dollars and a reduction of 2% to 5% in gross domestic products worldwide [6,7].

Antibiotics are used to treat diseases and infections in livestock animals as well as in humans. Antimicrobials are used for metaphylactic purposes and are given to the flocks or herds of animals at risk of diseases when some already have the clinical signs of infection for use as a nontherapeutic prevention of diseases. In some animals, e.g., chickens, cows, pigs, etc., they are also used as a growth promoter to increase the herd’s productivity. However, the usage of growth promoters was banned in the EU in 2006 due to the increasing concern of AMR [8].

Resistance bacteria can be transmitted directly from animals on farms to humans; through the food chain with the consumption of raw foods such as vegetables, fruits, ready-to-eat food (REF), etc.; or possibly through the consumption of inadequately cooked food, by cross-contamination with other food, or indirectly through the environment. Widespread use of antimicrobials in animals has led to the increased value of AMR, which potentially affects humans. The relationship of AMR transfers from animals to humans through the food chain is illustrated in Figure 1.

Our study aims to identify the potential transmission of antimicrobial-resistant microbes between humans and animals. Even though no formal technical assessment of the sample was conducted, the wide range of data collection from different scientific journals using an integrative study approach has allowed us to produce a comprehensive comparison of AMR to help policymakers, practitioners, and researchers to combat AMR and produce the laws to prevent it.

## 2. Materials and Methods

### 2.1. The Explication of the Field of Research

#### 2.1.1. China

With a population of approximately 1.3 billion people, China is one of the world’s largest consumers of antimicrobials. In the past three decades, the Chinese economy has grown in certain aspects, due to which the agriculture system has changed. As a result, China has become one of the leading users of antibiotics in the world, out of which more than half of the total consumption is in animals. A high rate of antibiotic usage has increased the growth of antibiotic-resistant bacteria transmitted between animals and humans directly or indirectly through the environment.

The estimated use of antibiotics in livestock is alarming; in 2010, 227 million tons of antibiotics were used, which increased to 298 million tons in 2020 [9]. In recent times, the Chinese government has made certain efforts to control the use of antibiotics in animals, for example, through the use of prescribed drugs in animals, restricting the use of drugs such as cephalosporin in animals, creating a list of prohibited drugs to use in agriculture, making the farmers record the list of antibiotics used, etc. [10,11,12].

#### 2.1.2. USA

Infectious disease is currently the world’s second most significant cause of death, ranking third in developed countries. Antibiotics are often used when raising animals, with unsafe antibiotic residue levels remaining in the meat after the animal is harvested or after harvesting the agricultural products. Approximately 80% of antibiotics supplied in the United States are used in animals, directly and indirectly, affecting livestock and the food chain by causing antimicrobial resistance to some extent.

AMR-related deaths in the US have affected more than 2.8 million people and cause 35,000 deaths annually [13]. The Centers for Disease Control and Prevention (CDC) has categorized the top 18 drug-resistant threats to the United States based on the specific level of concern: urgent, serious, and concerning. To tackle the growth of AMR through the food sources, the US government has made certain efforts, e.g., monitoring of antibiotic usage and sales, resistance and management practices being increased at multiple points in the food production chain, mandating that any medicine used while curing the animal or heard should be maintained, etc. [14].

### 2.2. Food Categories

Sixteen categories of food samples were analyzed, which include: chicken, pork, egg, beef, duck, mutton, fish, turkey, ham, meat, seafood, vegetables, fruits, milk, ready-to-eat food (RTE), and others (meat barbecue, grilled fish, chicken barbecue, soup, and rice).

### 2.3. Data Extraction

A systematic search was conducted independently in Medline via PubMed, Google Scholar, and Web of Science from 2012 to December 2021, using Medical Subject Headings (MeSH) to retrieve the data, e.g., AMR spreading from different food sources, AMR spreading from different pathogens, multidrug resistance, drug susceptibility tests for isolated pathogen, assessment method for multiple pathogens, China, USA, predefined keywords, and Boolean operators (AND, OR, NOT, and AND NOT).

### 2.4. Search Strategy

The following criteria were used while extracting the data: (1) access to the full text and abstract of the article; (2) food pathogen prevalence and AMR reported; (3) mention of pathogen analysis method; (4) sample sources (vegetable food origins, animal origins, poultry, dairy products, environmental samples, food handlers, etc.); (5) the AMR assessment method, which includes different molecular techniques such as disc diffusion and minimum inhibitory concentration (MIC); and (6) sample size and susceptible/resistant organisms that are multidrug resistant (MDR). Only research papers/articles published after 2012 were selected to be considered to ensure that the comparison focused on the contextual literature that could showcase current resistance patterns, prevention measures, and animals infection rates. Full text articles of eligible papers were assessed only after clearing the predefined process of inclusion and exclusion of articles. A hand search was conducted to assess all the papers to decrease the deduplication of data.

### 2.5. Screening and Data Extraction Process

Papers were managed using Mendeley (version 1.19.8, Elsevier, London, UK)), and the data from eligible papers were extracted independently using a standardized data extraction spreadsheet in Excel^®^ (Microsoft^®^ Office Excel 2013, Microsoft, Washington, DC, USA). Relevant data from papers included microbes, categories of food, food production, and food analysis. The number of samples with their phenotypic data related to AMR is represented in Table 1 and Table 2 and graphically represented in the form of a pie chart.

### 2.6. Data Analysis

A detailed analysis was first performed on all studies suiting the inclusion criteria, including those based on small sample sizes, in order be aware of quantifiable risks. Data were further described by food categories, corresponding bacteria species, and observed phenotypic and genotypic resistances.

## 3. Results

### 3.1. Descriptive Analysis of All Included Studies: General Findings

Data from 26 out of 132 research papers were selected for further examination. The selected papers are from different research journals which have collected samples from different parts of China and the USA, as shown in Figure 2. The 26 studies show food products from different sources, which are incorporated in Table 1 and Table 2.

Out of 26 papers selected, 15 papers were from China [15,16,17,18,19,20,21,22,23,24,25,26,27,28,29] and the rest (11 papers) were from the USA [30,31,32,33,34,35,36,37,38,39,40]. The data extracted contain 13,018 food samples, out of which 5000 samples were AMR positive (2276 and 2724 from China and the USA, respectively).

#### 3.1.1. China

Most of the microbial samples were from meat: out of 1691 total samples, 1022 were contaminated with microbes (chicken, pork, duck, beef, and mutton). A total of 639 out of 2362 infected microbial species were from aquatic products (fish and sea food), and 101 out of 216 infected microbial species were from milk, followed by fruits and vegetables, eggs, and ready-to-eat food (REF), which account for 329 out of 1094, 55 out of 847, and 32 out of 622 samples contaminated with microbes, respectively (Table 1).

Out of 6965 food samples, 2276 (32.7%) were contaminated with bacteria. *Salmonella* was seen in most of the positive samples, accounting for 9.9% (691) of the total positivity rate, followed by *V. parahaemolyticus*, in which 569 (8.2%) samples were positive. This was followed by *Enterococci* with 349 (5%), *E. coli* with 294 (4.2%), *Cronobacter* with 122 (1.8%), *C. coli* with 98 (1.4%), *S. aureus* with 90 (1.3%) and *C. jejuni* with 63 (0.9%) samples (Figure 3).

#### 3.1.2. USA

Most of the samples contaminated are from beef, chicken, and pork, in which 1036, 593, and 531 samples out of 1253, 1564, and 1461 were contaminated with microbes, respectively. This is followed by meat (veal and red meat), in which 188 samples were contaminated out of 396 samples; turkey, in which 128 samples were contaminated out of 299 samples; milk, in which 143 samples were contaminated out of 465 total samples; and others (meat barbecue, grilled fish, chicken barbecue, soup, and rice), vegetables, and fish, which account for 46 out of 366, 57 out of 194, and 2 out of 55 samples that were contaminated with microbes, respectively (Table 2).

Out of 6053 samples, 2274 (45.0%) samples were positive. *Staphylococcus aureus* was seen in most of the positive samples, accounting for 2111 (34.9%) of the total, followed by *Campylobacter* spp., in which 155 (2.6%) samples were positive. This was followed by *Enterococcus* with 121 (2.0%), *Staphylococcus aureus* MRSA with 114 (1.9%), *E. coli* with 72 (1.2%), *C. jejuni* with 69 (1.1%), *C. coli* with 66 (1.1%), *Listeria* spp. with 11 (0.2), *L. monocytogenes* with 3 (0.0%), and *Salmonella* with 2 (0.0%) samples (Figure 4).

When we compare the antimicrobial-resistant bacteria (AMRB) prevalence in both countries, we found that out of 13,018 food samples compiled, 4292 AMR isolates were yielded (2676 and 1616 were from China and the USA, respectively) (Figure 5). The majority of AMR isolate samples in China were from *Vibrio parahaemolyticus* (≤29%), which originated from aquatic products (fish and sea food), whereas in the USA, the majority of isolates were of *Staphylococcus aureus* that originated from meat products (beef, pork, chicken, turkey, and meat), vegetables, fish, and others, yielding ≤61% of the samples. This was followed by *Salmonella* (26%), *Enterococci* (14%), and *E. coli* (11%), and other bacteria that were ≥10% (*Cronobacter* spp. (6%), *C. coli* (6%), *C. Jejuni* (6%), and *Staphylococcus aureus* (4%)) in China, whereas in the USA, the AMR isolates were followed by *Campylobacter* (21%) and others that were ≥10% (*Enterococcus* (8%), *Staphylococcus aureus* (5%), *E. Coli* (4%), and *Listeria* spp. (1%)) (Figure 5).

### 3.2. Major Bacteria Groups and Their Relevant Food Product Categories with AMR

To better understand the distribution of AMR, we looked at the main species among foodborne pathogens and indicator bacteria independently from China and the USA (Figure 6 and Figure 7). β-lactam, and tetracycline resistance was the most common phenotypic AMR category across all groups (Figure 6 and Figure 7), followed by amino glycoside and quinolone resistance. Gram-positive bacteria were more likely to be resistant to tetracycline and macrolides, whereas Gram-negative bacteria were more likely to be resistant to aminoglycosides and quinolones.

### 3.3. Gram-Positive Bacteria

#### 3.3.1. *Enterococcus*

*Enterococcus faecium* and *Enterococcus faecalis* belong to the genus *Enterococcus*, which covers a wide range of food-related bacteria. The microorganisms are derived from animal products, vegetables, fruits, and fish-based foods. This includes raw, sliced, or whole fish; meat; and contaminated milk. These food items were procured from South East Asia and the United States of America. AMR was prevalent for tetracyclines, aminoglycosides, β-lactam, glycopeptides, and macrolides, with phosphonic antibiotics and fluoroquinolones being less prevalent (Figure 6 and Figure 7).

#### 3.3.2. *Staphylococcus*

Among other species, *S. aureus* was the most eminent member of its species, having two strains isolated: MRSA and MSSA. The prevalence of bacteria was commonly seen in food derived from animals such as the meat of chicken, pork, beef, chicken liver, and gizzards, as well as milk-derived products. The highest number of contaminated food samples was derived from the USA. AMR was most prevalent for β-lactam, macrolides, aminoglycosides, tetracyclines, and fluoroquinolones, followed by phenicol, c-phems, sulfonamides, glycopeptides, rifamycin, ansamycins, and folate inhibitors, which were less prevalent (Figure 6 and Figure 7).

#### 3.3.3. *Listeria* spp.

*L. monocytogenes*, *L. ivanovii*, *L. innocua*, and *L. welshimeri* belong to the genus *Listeria*. Out of these species, L. monocytogenes was most commonly seen in food. The infected food included vegetables such as onion, cucumber, peach, rutabaga, turnip, etc. The food products were procured from various western countries and also from the USA. AMR was most prevalent for β-lactam, tetracycline, and quinolones, with tetracycline, phenicol, and carbapenems being least prevalent (Figure 6 and Figure 7).

### 3.4. Gram-Negative Bacteria

#### 3.4.1. *Campylobacter*

*C. coli* and *C. jejuni* were the species found in the genus *Campylobacter*. The prevalence of bacteria in food categories was seen in chicken, milk, and vegetables. The food products were obtained from South East Asia and western countries, including China and the USA. AMR was observed against quinolones, tetracyclines, aminoglycosides, lincosamides, and macrolides, whereas in amphenicols and lincosamides, it was less prevalent (Figure 6 and Figure 7).

#### 3.4.2. *Escherichia coli*

*E. coli* was seen in most of the research papers. The majority of food products were meat, which included chicken, pork, beef, and mutton, and the other food categories were eggs, duck, fish, seafood, vegetables, fruits, milk, and REF. Most samples were obtained from China, which included most of the food products. AMR was observed against quinolones, phenicol, aminoglycosides, and β-lactam, whereas tetracyclines and c-phem were less prevalent (Figure 6 and Figure 7).

#### 3.4.3. *Salmonella*

*Salmonella enterica* and *Salmonella enteritidis* were the species found in the genus *salmonella*. The species contain various serovars: Enteritidis, Typhimurium, Ouakam, Typhi, Paratyphi, and Goldcoast. Most of the food products were chicken, pork, eggs, and vegetables. The majority of the samples were obtained from China. The AMR was observed against sulfonamides, quinolones, tetracycline, phenicol, β-lactam, and aminoglycosides, whereas folate pathway inhibitors and fluoroquinolones were less prevalent (Figure 6 and Figure 7).

#### 3.4.4. *Cronobacter*

The genus *Cronobacter* was represented by *C. sakazakii and C. malonaticus*. It was exclusively isolated from vegetables produced in China. Predominant isolates were related to cephalosporin and were less prominent in tetracycline and sulfonamides (Figure 6 and Figure 7).

#### 3.4.5. *Vibrio* spp.

*Vibrio parahaemolyticus*, *Vibrio cholera*, and *Vibrio alginolyticus* were the species found of the genus *Vibrio*. Most food products with these isolates were sea food, fish, and REF. These food items were procured from China. AMR was most prevalent for β-lactam, aminoglycosides, and cephalosporins, followed by quinolones, folate pathway inhibitors, tetracycline, and glycopeptides (Figure 6 and Figure 7).

## 4. Discussion

In this systematic data analysis, we intended to estimate the AMR exposure of users based on food from butcher shops and retail food markets in China and the USA. The study is intended to provide a literature-based analysis of AMR bacteria occurrence and vulnerability from food. The AMR exposure evaluation constitutes a basis for constructing an action plan against AMR, incorporating risk analysis into systemic surveillance systems at the consumer and retail level.

Quantitative AMR exposure was high for meat products with respect to foodborne pathogens such as *Salmonella* and *Staphylococcus aureus*, and also indicator AMRB such as *E. coli*, which shows resistance to high-priority, critically important AMs such as quinolones, cephalosporins (third class and higher generations), macrolides and ketolides, glycopeptides, and polymixins [41]. In this study, *Salmonella* was found to have the highest AMR against sulfonamides, quinolones, tetracyclines, and phenicols among the Gram-negative foodborne pathogens, with the AMR profile dependent on serovars. For the Gram-positive foodborne pathogens, *Staphylococcus aureus* shows the highest AMR against β-lactam, macrolides, and aminoglycosides. These antimicrobial-resistant bacteria pathogens can be transmitted between food processing, animals, and consumers through direct or indirect contact (e.g., transfer of livestock associated with *Staphylococcus aureus* MRSA) [42]. The indicator bacteria *E.coli* shows AMR against β-lactam, aminoglycosides, and tetracyclines. The systematic data of food from the retail market matches the key AMR as reported for China and the USA [43]. However, the existing reports that are available mainly focus on Gram-negative and Gram-positive foodborne bacteria in meat products. Due to this, there is only a limited understanding of AMRB in other food categories, as well as AMR in technologically important bacteria, including starter cultures. As a result, this study provides a systematic overview and estimation of AMR exposure data for food at retail, covering a wide range of food categories and microbe groups, as a precursor for developing a comprehensive AMR risk assessment for the consumer.

Consumers face a multifaceted risk, influenced by the preparation of food and cooking habits. Because most raw meat products are cooked before consumption, which reduces the number of bacteria, the final AMRB exposure level can vary greatly depending on hygiene strategies as inadequate and irregular hygiene practices may increase foodborne illness [44,45,46]. *Salmonella* and *Staphylococcus aureus* are the predominating foodborne pathogens worldwide, causing 1.35 million and 2.41 million illnesses yearly, respectively [47,48]. Most of the cases of foodborne illness are due to cross-contamination because in China and other Western countries, sharing plates or cutlery for raw and cooked meat can result in cross-contamination from raw chicken to cooked food. The consumption of raw meat in various Chinese recipes is a major risk factor for food-related illness each year [49]. This suggests that proper hygiene practices are not being followed in a systematic manner. Raw meat products are, thus, a major factor in the cross-contamination of bacteria in the kitchen or at the table that transmits infection, and thus, are likely also involved in the transmission of AMRB to people. In contrast to raw meat or other edible items that undergo the cooking step prior to eating, ready-to-eat food, fruits, and some vegetables are consumed without the cooking step, which resulted in a high level of colony counts of relevant bacteria and indicator bacteria. As a result, if AMRB is available, it may transmit in considerably higher numbers from food to the consumer [50]. Because limited data prevented further exposure calculations for other REF products and starter cultures, the qualitative analysis in this study revealed that AMRB is present in many food products, and require increased systematic surveillance. In order to estimate the associated public health risk, this surveillance scheme should include determining the potential for AMR gene transfer between starter culture bacteria, such as *E.coli*, as well as to gut commensals, an obligate and opportunistic pathogens. Therefore, studying the genetic organization of AMR genes is critical for assessing AMR risk and should be conducted more systematically. The different approaches taken by the countries to safeguard the health of people have been summarized in Table 3.

On a descriptive basis, these potential knowledge gaps were supported by indications of AMRB exposure from food categories with insufficient data tables. We tried to incorporate the maximum categories of food products in our dataset, such as ready-to-eat food (fermented food, rice bowls, etc.), veal, and dairy products such as milk, cheese, yogurt, etc. A descriptive analysis of these food categories revealed a high level of AMR. The systematic surveillance data retrieved in this systematic meta-analysis is insufficient to correctly assess the risk of AMR exposure through these food items, yet it will be helpful to analyze the foodborne pathogens from various sources and decrease the knowledge gap to prevent foodborne-related hospitalization.

## 5. Conclusions

The transfer of AMR from various food products to consumers may occur via multiple recognized mechanisms. The processing of raw food in retail marketplaces is more common and yet difficult to monitor. Surveillance is carried out on farms by research organizations, and government authorities frequently test AMR levels in food processing facilities and on items sold in retail markets. Although it may not be feasible to obtain statistics that represent customers’ real food handling practices, initiatives are ongoing at many levels to assist users in better understanding how food is produced and what they can do to lower their risk of contracting AMR illnesses from food items.

This systematic meta-analysis shows the current condition of AMR in edible products in the food market, as well as the recognition of knowledge gaps and several government organizations related to AMR in food. The maximum level of AMRB exposure was found in raw meat, aquatic products, and dairy products for Gram-positive *Enterococci*, *Staphylococcus aureus*, and *Listeria* spp. and Gram-negative bacteria such as *Escherichia coli, Vibrio, Cronobacter, Salmonella, Campylobacter,* etc., which showed AMR against β-lactam, tetracyclines, aminoglycosides, quinolones, fluoroquinolones, macrolides, tetracycline, phenicol, and c-phem.

Given the large amount of research within the field, the systematization and evaluation of existing results can aid in providing an accurate interpretation of the existing data. This meta-analysis was not intended to establish epidemiological connections between the existence of resistant AMRB isolated from various food sources and the development of resistance in humans. Rather, it was used as a powerful tool for summarizing and comparing findings across a wide range of primary studies. Nevertheless, it is likely that the risk of transmitting AMRB to humans can be reduced if proper food processing techniques are followed, and hygiene standards are maintained by the consumers in the kitchen.

Nonetheless, this review provided critical information on AMRB exposure in retail food for the design of future AMR risk evaluations based on actual China and USA data, and it will aid in optimizing AMR monitoring schemes. This is especially pertinent to: (i) the implementation of systematic AMR monitoring schemes for food at retail, (ii) the inclusion of additional food categories such as ready-to-eat and novel food products, and (iii) the exploration of AMR genetics. As an additional core pillar of a One Health strategy for AMR monitoring and response systems, we recommend designing and enforcing systematic phenotypic and genotypic surveillance of AMR in retail food to minimize the knowledge gap among consumers.

## Figures and Tables

**Figure 1 antibiotics-11-01471-f001:**
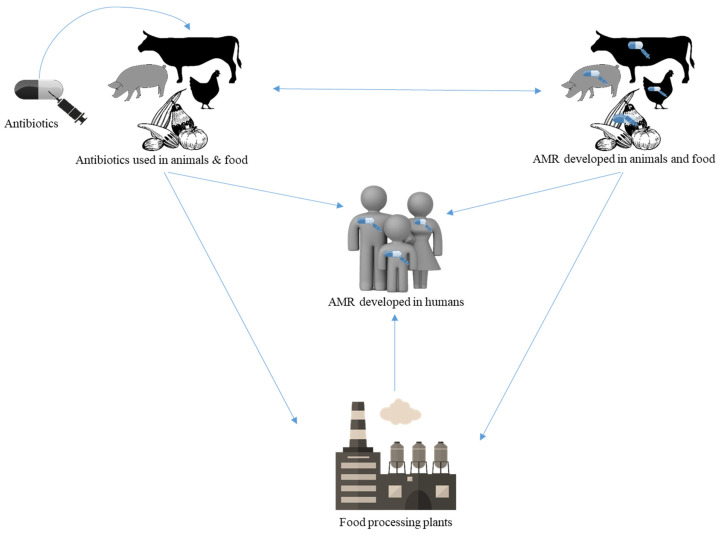
Relationship between antibiotic usage and antimicrobial resistance development in animals and humans through food chain.

**Figure 2 antibiotics-11-01471-f002:**
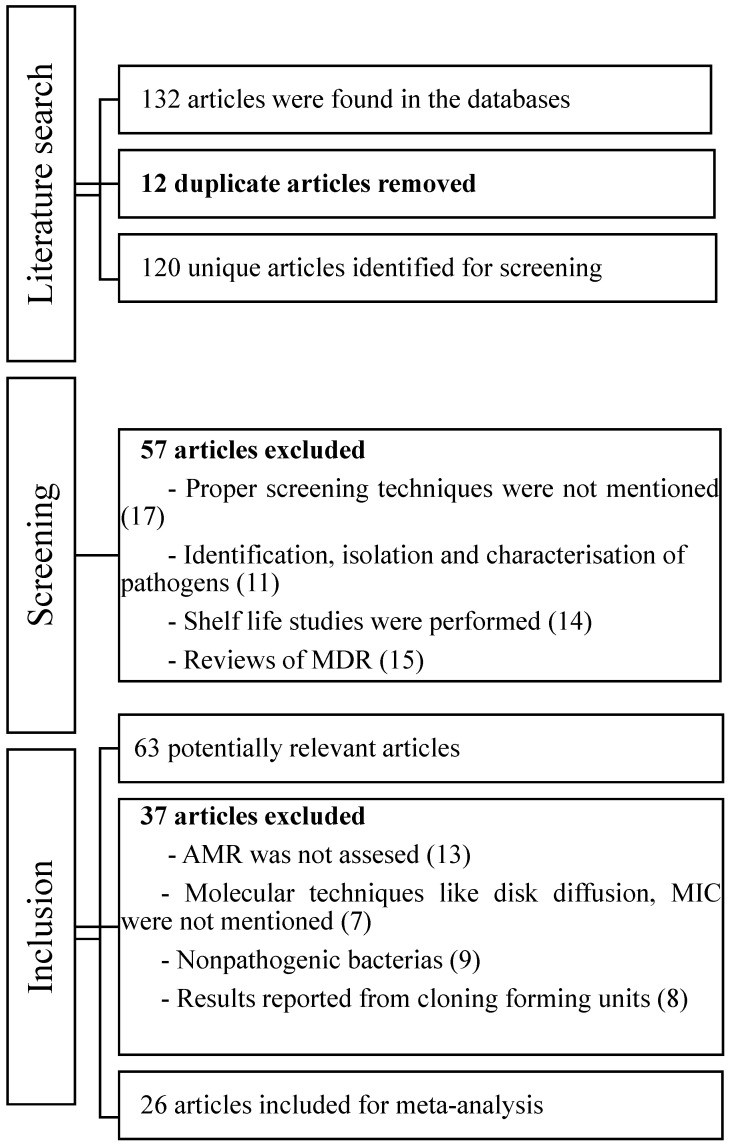
A strategy for identifying, screening, and including articles in meta-analyses from different food sources and their AMR.

**Figure 3 antibiotics-11-01471-f003:**
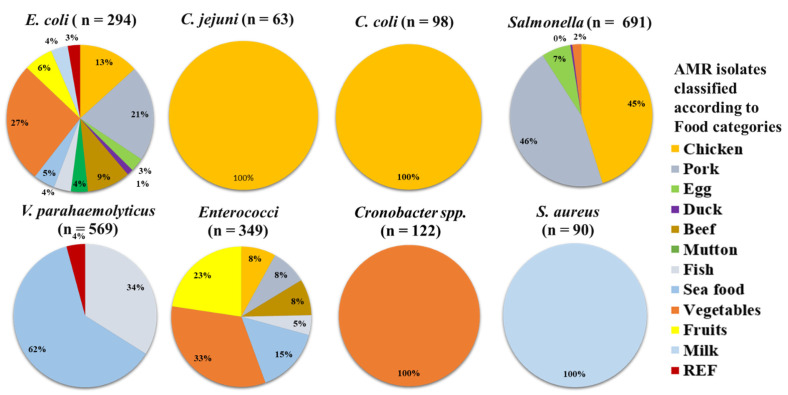
Distribution of the number of food samples in China (n = 2276) that had AMR isolates in them, broken down by food types. In each pie chart, the total at the top indicates the (n) number of samples used to produce the graph.

**Figure 4 antibiotics-11-01471-f004:**
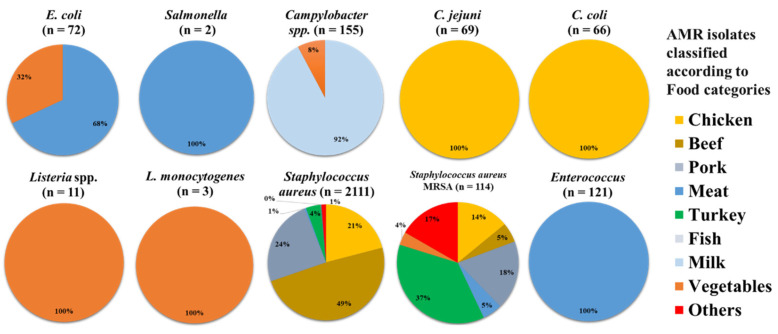
Distribution of the number of food samples in the USA (n = 2724) that had AMR isolates in them, broken down by food types. In each pie chart, the total at the top indicates the (n) number of samples used to produce the graph.

**Figure 5 antibiotics-11-01471-f005:**
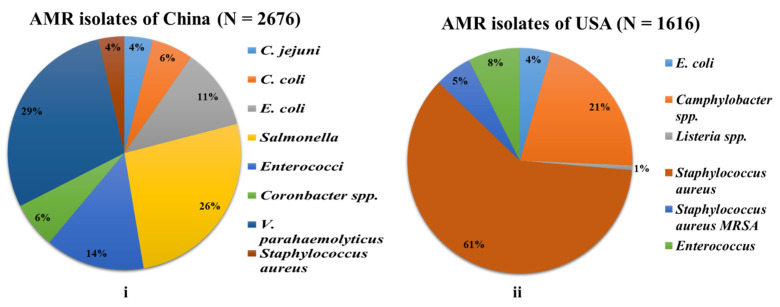
Individual AMR isolates with their respective percentages: (**i**) China; (**ii**) USA.

**Figure 6 antibiotics-11-01471-f006:**
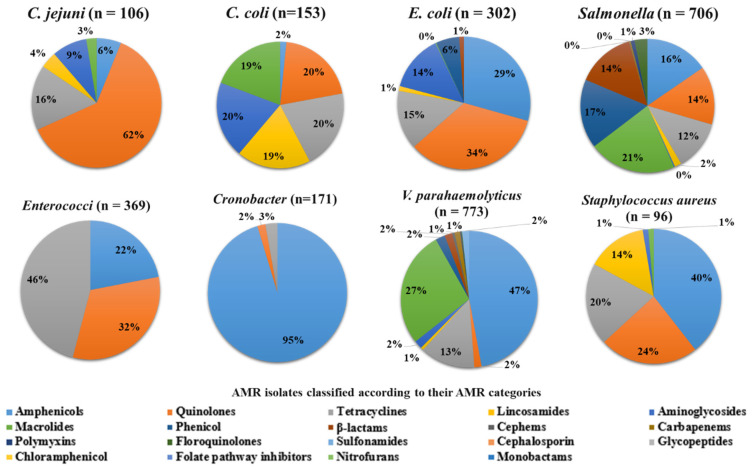
Based on the phenotypic AMR detected against the major AM classes in China, distributions of AMR isolates of foodborne pathogens and indicator bacteria have been determined (n = 2676) (n = total number of food samples contaminated with certain bacteria).

**Figure 7 antibiotics-11-01471-f007:**
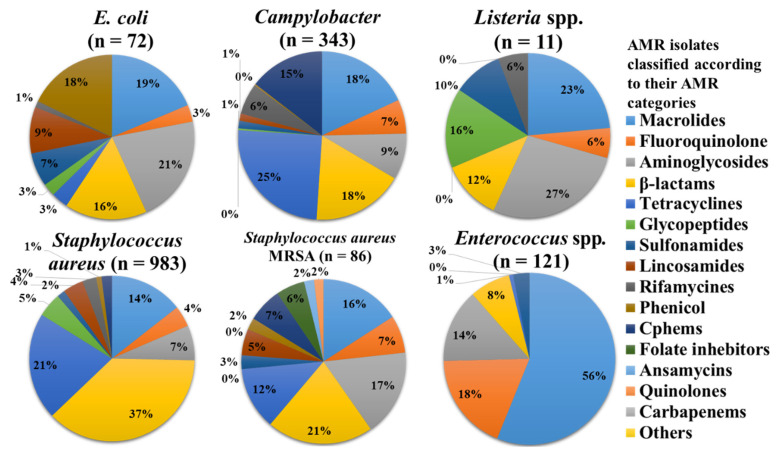
Based on the phenotypic AMR detected against the 16 major AM classes in the USA, distributions of AMR isolates of foodborne pathogens and indicator bacteria have been determined (n = 1616). (n = total number of food samples contaminated with certain bacteria).

**Table 1 antibiotics-11-01471-t001:** Food product categories along with microbes that have been detected in China from 2012 to 2021, where n = total number of food samples contaminated; % of isolates.

Microbes	Chicken n = 1006	Pork n = 558	Egg n = 847	Duck n = 41	Beef n = 67	Mutton n = 19	Fish n = 1108	Sea Food n = 1254	Vegetables n = 1094	Fruits n = 132	Milk n = 216	REF n = 622	Total n = 6965
	n and (%)	n and (%)	n and (%)	n and (%)	n and (%)	n and (%)	n and (%)	n and (%)	n and (%)	n and (%)	n and (%)	n and (%)	n and (%)
*E. coli*	39 (3.9)	62 (11.1)	9 (1.1)	4 (9.8)	28 (41.8)	11 (57.9)	11 (1.0)	14 (1.1)	78 (7.1)	19 (14.4)	11 (5.1)	8 (1.3)	294 (4.2)
*C. jejuni*	63 (6.3)	-	-	-	-	-	-	-	-	-	-	-	63 (0.9)
*C. coli*	98 (9.7)	-	-	-	-	-	-	-	-	-	-	-	98 (1.4)
*Salmonella*	312 (31)	316 (56.6)	46 (5.4)	3 (7.3)	-	-	-	-	14 (1.3)	-	-	-	691 (9.9)
*V. parahaemolyticus*	-	-	-	-	-	-	193 (17.4)	352 (28.1)	-	-	-	24 (3.9)	569 (8.2)
*Enterococci*	28 (2.8)	29 (5.2)	-	-	29 (43.3)	-	16 (1.4)	53 (4.2)	115 (10.5)	79 (59.8)	-	-	349 (5.0)
*Cronobacter* spp.	-	-	-	-	-	-	-	-	122 (11.2)	-	-	-	122 (1.8)
*S. aureus*	-	-	-	-	-	-	-	-	-	-	90 (41.7)	-	90 (1.3)
Total infected	540 (53.7)	407 (72.9)	55 (6.5)	7 (17.1)	57 (85.1)	11 (57.9)	220 (19.9)	419 (33.4)	329 (30.1)	98 (74.2)	101 (46.8)	32 (5.1)	2276 (32.7)

**Table 2 antibiotics-11-01471-t002:** Food product categories along with microbes that have been detected in USA from 2012 to 2021, where n = total number of food samples contaminated; % of isolates.

Microbes	Chicken n = 1564	Beef n= 1253	Pork n = 1461	Meat n = 396	Turkey n = 299	Fish n = 55	Milk n = 465	Vegetables n = 194	Others n = 366	Total n = 6053
	n and (%)	n and (%)	n and (%)	n and (%)	n and (%)	n and (%)	n and (%)	n and (%)	n and (%)	n and (%)
*E. coli*	-	-	-	49 (12.4)	-	-	-	23 (11.9)	-	72 (1.2)
*Salmonella*	-	-	-	2 (0.5)	-	-	-	-	-	2 (0.0)
*Campylobacter* spp.	-	-	-	-	-	-	143 (30.8)	12 (6.2)	-	155 (2.6)
*C. jejuni*	69 (4.4)	-	-	-	-	-	-	-	-	69 (1.1)
*C. coli*	66 (4.2)	-	-	-	-	-	-	-	-	66 (1.1)
*Listeria* spp.	-	-	-	-	-	-	-	11 (5.7)	-	11 (0.2)
*L. monocytogenes*	-	-	-	-	-	-	-	3 (1.5)	-	3 (0.0)
*Staphylococcus aureus*	442 (28.3)	1030 (82.2)	510 (34.9)	10 (2.5)	86 (28.8)	2 (3.6)	-	4 (2.1)	27 (7.4)	2111 (34.9)
*Staphylococcus aureus* MRSA	16 (1.0)	6 (0.5)	21 (1.4)	6 (1.5)	42 (14.0)	-	-	4 (2.1)	19 (5.2)	114 (1.9)
*Enterococcus*	-	-	-	121 (30.5)	-	-	-	-	-	121 (2.0)
Total infected	593 (37.9)	1036 (82.7)	531 (36.3)	188 (47.5)	128 (42.8)	2 (3.6)	143 (30.8)	57 (29.4)	46 (12.6)	2724 (45.0)

**Table 3 antibiotics-11-01471-t003:** Surveillance bodies formed by China and USA to control AMR.

S. No.	Country	Organizations	Role
1.	China	Bureau of Animal and Plant Health Inspection and Quarantine (BAPHIQ) https://www.baphiq.gov.tw/ (accessed on 24 April 2022)	Global Action Plan on Antimicrobial Resistance, and the OIE Strategy on Antimicrobial Resistance and the Prudent Use of Antimicrobials.
2.		National Action Plan (NAP) http://www.gov.cn/xinwen/2016-08/25/content_5102348.htm (accessed on 24 April 2022)	Regulate antimicrobial agents and antimicrobial resistance control.
3.		China Antimicrobial Resistance Surveillance System (CARSS) http://www.carss.cn/ (accessed on 24 April 2022)	AMR surveillance.
4.		China Antimicrobial Surveillance Network (CHINET) https://www.chinets.com/ (accessed on 24 April 2022)	Help clinicians to better understand the current status and trends of AMR and to correct inappropriate antibiotic prescribing.
5.	USA	National Antimicrobial Resistance Monitoring System (NARMS) https://www.cdc.gov/narms/index.html (accessed on 24 April 2022)	Track changes in the antimicrobial susceptibility of enteric (intestinal) bacteria found in ill people.
6.		Centers for Disease Control and Prevention https://www.cdc.gov/ (accessed on 24 April 2022)	Carry out scientific research on new and ongoing pathogen threats.
7.		Food and Drug Administration (FDA) https://www.fda.gov/ (accessed on 24 April 2022)	Protecting public health by assuring that foods are wholesome, sanitary, and properly labeled.
8.		US Department of Agriculture (USDA) https://www.usda.gov/ (accessed on 24 April 2022)	Safeguard food, agriculture, natural resources, rural development, nutrition, and related issues based on public policy.

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
