# Peer review of "Systematic Surveillance and Meta-Analysis of Antimicrobial Resistance and Food Sources from China and the USA"

_antibiotics, 2022, doi:10.3390/antibiotics11111471_

Round 1

Reviewer 1 Report (New Reviewer)

This is a systematic review and meta-analysis of antibiotics resistance involved in food products from two countries. Please find my comments. First, I don’t like a title, in my opinion it not fully reflects a content of this work.

Page 13 please explain authors statement “ because in China and other Western countries were shared…”

Not all food categories listened in Materials and Methods were shown in table 1 and 2.

In addition, the different layout of the tables makes comparison between results from both countries difficult.

In order to increase the attractiveness of the article to the reader I suggest addition of some comparison between results obtained in two countries (in graphs if possible) together with some explanation of observed differences.

Author Response

Thank you so much for your positive feedback.

Reviewer 2 Report (New Reviewer)

I understand the authors have done a lot of efforts to search for scientific literature, but the presentation is relatively poor. I have many concerns that are detailed in the attachment. I hope authors reconsider to draft the manuscript as a review article, rather than a research article. I have detected redundant results (same content but presented in both one figure and one table). I also noted that the citations are not properly cited. The methods are also not convincing (especially the search criteria).

Author Response

Thank you for your valuable suggestions. We attempted to represent food product categories as well as microbes in the table, and we had the distribution of the number of food samples that had AMR isolates in them, broken down by food type, in the pie chart, to facilitate comparisons between the two countries in the article. The citations have been added to the updated manuscript. 

The suggested changes have been corrected in the updated manuscript including more details about the methodology.

Reviewer 3 Report (New Reviewer)

General comment: The research article entitled “Systematic Surveillance and Meta-analysis of Antimicrobial Resistance and Food Sources from China and USA” is a well-organized study, with sufficient methodology and adequate description of the results. Some minor corrections are required for the improvement of the manuscript.

Abstract: The Abstract is well written and adequately presents the aim and the basic results of the study.

- Authors could add  more data about the methodology used.

Introduction: The introduction section adequately covers the basic aim of the study.

Materials and Methods:  The materials and methods are adequately presented.

-Could authors add more details about the methodology used for the metanalysis and statistical analysis process that used?

Results: The results of the study are analytically presented. Tables and Figures are adequate explain the findings of the study.

Conclusion: The conclusion is adequate and summarizes the main text.

Bibliography/References: The references used by the authors cover adequately the relative scientific field and the aims of the study.

Author Response

Thank you so much for your valuable suggestions and positive feedback.  We have added more details about the methodology in the updated manuscript.

This manuscript is a resubmission of an earlier submission. The following is a list of the peer review reports and author responses from that submission.

Round 1

Reviewer 1 Report

Dear author,

The text needs extensive proofreading in English and spelling.

In the title did you mean Systemic or Systematic?

Why a study including China and the US? My suggestion is to rewrite with well-established parameters and choose just one country so that the data can be better analyzed and explored.

The proposal to carry out a meta-analysis and systematic review was not achieved. Parameters were lacking to describe the analysis process. Statistical analysis of data processing was not performed and complementary research data were not provided. When thinking about doing a systematic review, it is necessary to clearly present the data analysis and a check of data consistency. In the proposed study, the parameters used, the exclusion parameters and the analyzes obtained were not clear.

Also, the written form is unclear. There are spelling errors, the references are not standardized, the writing of the mentioned bacterial genera and species also have errors (i.e. Cronobacter spp., among others). Figure 1. Does not add anything to the search. The data presented in the graphs was not the best form of presentation. The conclusion needs to be improved, it does not clearly reflect the two countries studied.

Unfortunately, in light of the observations I propose to reject the manuscript.

Reviewer 2 Report

The manuscript aims to describe antimicrobial resistance associated with different food categories in USA and China in the period between 2012 and 2021. The article could be interesting, showing antimicrobial resistance trends and evolution associated with different bacteria of food origin in the last ten years. Additionally, it could be helpful to better understand AMR epidemiology associated with the food chain, identifying potential food products with high AMR risk.

The manuscript is overall disorganised, rough and difficult to understand.

The introduction briefly describes AMR mechanisms, causes and costs, but should be more focused on AMR associated with food products, to support and explain the author's choice of focusing only on this aspect of AMR. For example, it could be helpful to deepen pathways connecting AMR bacteria and the food chain, outbreaks associated with consumption of food contaminated with AMR bacteria, the role of food in the dissemination of bacteria with CIA resistance etc..

In the “Explication of field of research”, paragraphs 2.1.1 and 2.1.2 are too ambiguous. It is unclear why the authors have been focused on China and USA: have these countries the highest livestock population? the highest antimicrobial consumption? Are frequently foodborne outbreaks described?

Paragraph 2.2: What does “Others” category include?

Pararagraph 2.3, 2.4, 2.5 describe the selection process and the analysis of manuscripts included in the study. I suggest entitling “Search strategy”, “Data extraction” and “Data analysis” the 3 paragraphs and clearly describe the manuscript selection process and the analysis performed.

In paragraph 2.4, the authors describe the relevant data considered in the article, including “bacteria isolated from food sources, food categories” etc. What do the authors mean by “their production” referred to “food category”?

In paragraph 2.5 the authors referred to “all studies suiting with the inclusion criteria, involving those based on small sample sizes”. Is sample size another criterion considered in the manuscript selection process?

The result section is chaotic.

I suggest to divide the result section in at least 4 paragraphs:

1: A general description of the studies considered in the review, including number of studies, origin (USA, China), year of the study, food source, reference of the study. Data could be represented in a table.

2, 3: For both China and USA, a general description of food sources, number of food samples associated with AMR bacteria, AMR bacteria species, phenotypic and genotypic AMR profile. Main considerations should be described in the text, meanwhile other data could be represented in a table. For example, most data in paragraphs 3.2 and 3.3 could be summarised in a table and briefly commented in the text. At the same time, data in Figure 2,3,4 could be better represented in a table.

Specific comments:

-        in paragraph 3.1.1 and 3.1.2: it is unclear what does “positive sample” mean and when authors refer to samples associated with isolation of bacteria or the presence of AMR bacteria, both in text and figure/table. For example: Table 1 refers to microbes that have been detected in China, meanwhile Figure 2 refers to “Distribution of the number of food samples in China (N = 2276) that had AMR isolates in them”. However, the number of samples are the same in both Table 1 and Figure 2.

-        there is inconsistency in the number of AMR isolates considered in the study. Are they 5000 (Abstract and paragraph 3.1) or 4292 (Paragraph 3.1.2)?

-        genotypic AMR data are not described in the result section.

4: Comparison of USA and China AMR data, also with statistical analysis.

An additional paragraph could be focused on the evolution of AMR phenotypic/genotypic profile and the association with food products in the last ten years.

The discussion appears sketchy and should be improved.

I suggest implementing the discussion focusing on AMR in China and USA and

-        explain if the most common AMR profiles are consistent with AMR stewardship in livestock in these countries,

-        identify food products associated with the highest AMR, possible reasons and establish if these food products have been involved in human outbreaks in the last ten years;

-        verify how antimicrobial resistance profiles have been changed in the last ten years and if the strategies to reduce AMR have influenced this evolution.

Tables:

Please check all the tables.

For example, Table 1: in column "Duck" the percentage of "total infected" is 7/41*100 = 17.1%. There are discrepancies in number approximation: for example “total infected” of “Milk” is 101/216*100= 46.759 (approximated to 46.7) meanwhile “Total infected” of “Vegetables” is 329/1094*100 = 30.073 (approximated to 30.1).